# Non-decreasing Quantile Function Network with Efficient Exploration for Distributional Reinforcement Learning

## Abstract

Although distributional reinforcement learning (DRL) has been widely examined in the past few years, there are two open questions people are still trying to address. One is how to ensure the validity of the learned quantile function, the other is how to efficiently utilize the distribution information. This paper attempts to provide some new perspectives to encourage the future in-depth studies in these two fields. We first propose a non-decreasing quantile function network (NDQFN) to guarantee the monotonicity of the obtained quantile estimates and then design a general exploration framework called distributional prediction error (DPE) for DRL which utilizes the entire distribution of the quantile function. In this paper, we not only discuss the theoretical necessity of our method but also show the performance gain it achieves in practice by comparing with some competitors on Atari 2600 Games especially in some hard-explored games.

## 1 Introduction

Distributional reinforcement learning (DRL) algorithms (Jaquette, 1973; Sobel, 1982; White, 1988; Morimura et al., 2010; Bellemare et al., 2017), different from the value based methods (Watkins, 1989; Mnih et al., 2013) which focus on the expectation of the return, characterize the cumulative reward as a random variable and attempt to approximate its whole distribution. Most existing DRL methods fall into two main classes according to their ways to model the return distribution. One class, including categorical DQN (Bellemare et al., 2017), C51, and Rainbow (Hessel et al., 2018), assumes that all the possible returns are bounded in a known range and learns the probability of each value through interacting with the environment. Another class, for example the QR-DQN (Dabney et al., 2018b), tries to obtain the quantile estimates at some fixed locations by minimizing the Huber loss (Huber, 1992) of quantile regression (Koenker, 2005). To more precisely parameterize the entire distribution, some quantile value based methods, such as IQN (Dabney et al., 2018a) and FQF (Yang et al., 2019), are proposed to learn a continuous map from any quantile fraction $\tau \in (0, 1)$ to its estimate on the quantile curve.

The theoretical validity of QR-DQN (Dabney et al., 2018b), IQN (Dabney et al., 2018a) and FQF (Yang et al., 2019) heavily depends on a prerequisite that the approximated quantile curve is non-decreasing. Unfortunately, since no global constraint is imposed when simultaneously estimating the quantile values at multiple locations, the monotonicity can not be ensured by using any existing network design. At early training stage, the crossing issue is even more severe given limited training samples. Another problem to be solved is how to design an efficient exploration method for DRL. Most existing exploration techniques are originally designed for non-distributional RL (Bellemare et al., 2016; Ostrovski et al., 2017; Tang et al., 2017; Fox et al., 2018; Machado et al., 2018), and very few of them can work for DRL. Mavrin et al. (2019) proposes a DRL-based exploration method, DLTV, for QR-DQN by using the left truncated variance as the exploration bonus. However, this approach can not be directly applied to quantile value based algorithms since the original DLTV method requires all the quantile locations to be fixed while IQN or FQF resamples the quantile locations at each training iteration and the bonus term could be extremely unstable.

To address these two common issues in DRL studies, we propose a novel algorithm called Non-Decreasing Quantile Function Network (NDQFN), together with an efficient exploration strategy,

distributional prediction error (DPE), designed for DRL. The NDQFN architecture allows us to approximate the quantile distribution of the return by using a non-decreasing piece-wise linear function. The monotonicity of $N+1$ fixed quantile levels is ensured by an incremental structure, and the quantile value at any $\tau \in (0, 1)$ can be estimated as the weighted sum of its two nearest neighbors among the $N+1$ locations. DPE uses the 1-Wasserstein distance between the quantile distributions estimated by the target network and the predictor network as an additional bonus when selecting the optimal action. We describe the implementation details of NDQFN and DPE and examine their performance on Atari 2600 games. We compare NDQFN and DPE with some baseline methods such as IQN and DLTV, and show that the combination of the two can consistently achieve the optimal performance especially in some hard-explored games such as Venture, Montezuma Revenge and Private Eye.

For the rest of this paper, we first go through some background knowledge of distributional RL in Section 2. Then in Sections 3, we introduce NDQFN, DPE and describe their implementation details. Section 4 presents the experiment results using the Atari benchmark, investigating the empirical performance of NDQFN, DPE and their combination by comparing with some baseline methods.

## 2 BACKGROUND AND RELATED WORK

Following the standard reinforcement learning setting, the agent-environment interactions are modeled as a Markov Decision Process, or MDP, $(\mathcal{X}, \mathcal{A}, R, P, \gamma)$ (Puterman, 2014). $\mathcal{X}$ and $\mathcal{A}$ denote a finite set of states and a finite set of actions, respectively. $R : \mathcal{X} \times \mathcal{A} \rightarrow \mathbb{R}$ is the reward function, and $\gamma \in [0, 1)$ is a discounted factor. $P : \mathcal{X} \times \mathcal{A} \times \mathcal{X} \rightarrow [0, 1]$ is the transition kernel.

On the agent side, a stochastic policy $\pi$ maps state $x$ to a distribution over action $a \sim \pi(\cdot|x)$ regardless of the time step $t$. The discounted cumulative rewards is denoted by a random variable $Z^\pi(x, a) = \sum_{t=0}^\infty \gamma^t R(x_t, a_t)$, where $x_0 = x$, $a_0 = a$, $x_t \sim P(\cdot|x_{t-1}, a_{t-1})$ and $a_t \sim \pi(\cdot|x_t)$. The objective is to find an optimal policy $\pi^*$ to maximize the expectation of $Z^\pi(x, a)$, $\mathbb{E}Z^\pi(x, a)$, which is denoted by the state-action value function $Q^\pi(x, a)$. A common way to obtain $\pi^*$ is to find the unique fixed point $Q^* = Q^{\pi^*}$ of the Bellman optimality operator $\mathcal{T}$ (Bellman, 1966):

$$Q^\pi(x, a) = \mathcal{T}Q^\pi(x, a) := \mathbb{E}[R(x, a)] + \gamma \mathbb{E}_P \max_{a'} Q^*(x', a'). \tag{1}$$

The state-action value function $Q$ can be approximated by a parameterized function $Q_\theta$ (e.g. a neural network). Q -learning (Watkins, 1989) iteratively updates the network parameters by minimizing the squared temporal difference (TD) error

$$\delta_t^2 = \left[ r_t + \gamma \max_{a' \in \mathcal{A}} Q_\theta(x_{t+1}, a') - Q_\theta(x_t, a_t) \right]^2,$$

on a sampled transition $(x_t, a_t, r_t, x_{t+1})$, collected by running an $\epsilon$-greedy policy over $Q_\theta$.

Distributional RL algorithms, instead of directly estimating the mean $\mathbb{E}Z^\pi(x, a)$, focus on the distribution $Z^\pi(x, a)$ to sufficiently capture the intrinsic randomness. The distributional Bellman operator, which has a similar structure with (1), is defined as follows,

$$\mathcal{T}Z^\pi(x, a) \overset{D}{=} R(x, a) + \gamma Z^\pi(X', A'),$$

where $X' \sim P(\cdot|x, a)$ and $A' \sim \pi(\cdot|X')$. $A \overset{D}{=} B$ denotes the equality in probability laws.

## 3 NON-MONOTONICITY IN DISTIRBUTIONAL REINFORCEMENT LEARNING

In Distributional RL studies, people usually pay attention to the quantile function $F_Z^{-1}(\tau) = \inf\{z \in \mathbb{R} : \tau \leq F_Z(z)\}$ of total return $Z$, which is the the inverse of the cumulative distribution function $F_Z(z) = Pr(Z < z)$. In practice, with limited observations at each quantile level, it is much likely that the obtained quantile estimates $F_Z^{-1}(\tau)$ at multiple given locations $\{\tau_1, \tau_2, \ldots, \tau_N\}$ for some state-action pair $(x, a)$ are non-monotonic as Figure 1 (a) illustrates. The key reason behind this phenomenon is that the quantile values at multiple quantile fractions are separately estimated without applying any global constraint to ensure the monotonicity. Ignoring the non-decreasing

property of the learned quantile function leaves a theory-practice gap, which results in the potentially non-optimal action selection in practice. Although the crossing issue has been broadly studied by the statistics community (He, 1997; Chernozhukov et al., 2010; Dette & Volgushev, 2008), how to ensure the monotonicity of the approximated quantile function in DRL still remains challenging, especially to some quantile value based algorithms such as IQN and FQF, which do not focus on fixed quantile locations during the training process.

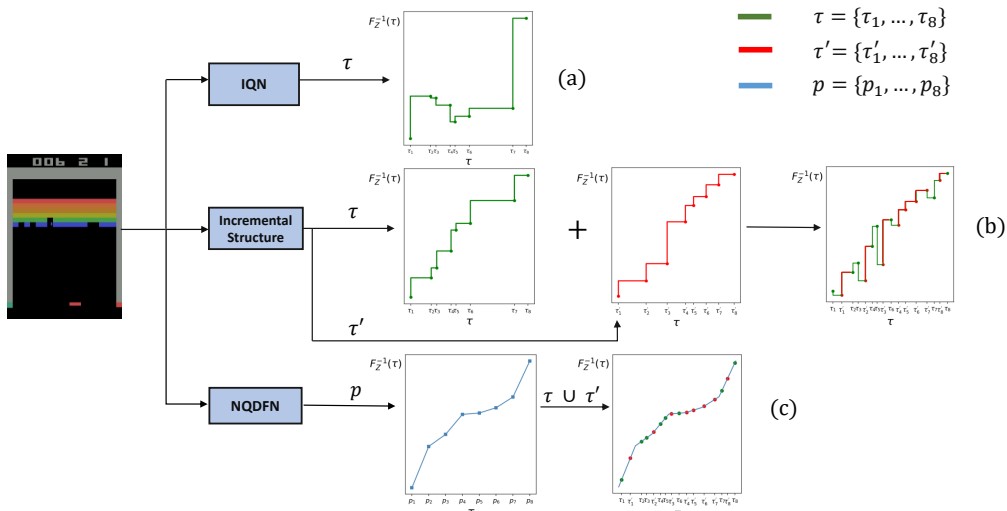

Figure 1: Quantile estimates obtained by (a) IQN, (b) mere incremental structure and (c) NDQFN under 50M training on Breakout. $\tau$ and $\tau'$ denote two sets of sampled quantile fractions from two different training iterations.

## 4 NON-DECREASING QUANTILE FUNCTION NETWORK

To address the crossing issue, we introduce a novel Non-Decreasing Quantile Function Network (NDQFN) with two main components: (1). an incremental structure to estimate the increase of quantile values between two pre-defined nearby supporting points $p_{i-1} \in (0,1)$ and $p_i \in (0,1)$, i.e., $\Delta_i(x, a; \boldsymbol{p}) = F_Z^{-1}(p_i) - F_Z^{-1}(p_{i-1})$, and subsequently obtain each $F_Z^{-1}(p_i)$ for $i \in \{0, \ldots, N\}$ as the cumulative sum of $\Delta_i's$; (2). a piece-wise linear function which connects the $N + 1$ supporting points to represent the quantile estimate $F_{Z(x,a)}^{-1}(\tau)$ at any given fraction $\tau \in (0,1)$. Figure 2 describes the whole architecture of NDQFN.

We first describe the incremental structure. Let $\boldsymbol{p} = \{p_0, \cdots, p_N\}$ be a set of $N + 1$ supporting points, satisfying $0 <\leq p_i \leq p_{i+1} < 1$ for each $i \in \{1, 2, \ldots, N-1\}$. Thus, $Z(x, a)$ can be parameterized by a mixture of $N$ Diracs, such that

$$Z_{\boldsymbol{\theta}, \boldsymbol{p}}(x, a) := \sum_{i=0}^{N-1} (p_{i+1} - p_i) \delta_{\theta_i(x,a)}, \tag{2}$$

where each $\theta_i$ denotes the quantile estimation at $\hat{p}_i = \frac{p_i + p_{i+1}}{2}$, and $\boldsymbol{\theta} = \{\theta_0, \ldots, \theta_{N-1}\}$. Let $\psi : \mathcal{X} \to \mathbb{R}^d$ and $\phi : [0,1] \to \mathbb{R}^d$ represent the embeddings of state $x$ and quantile fraction $\tau$, respectively. The baseline value $\Delta_0(x, a; \boldsymbol{p}) := F_{Z(x,a)}^{-1}(p_0)$ at $p_0$ and the following $N$ non-negative increments $\Delta_i(x, a; \boldsymbol{p})$ for $i \in \{1, \cdots, N\}$ can be represented by

$$\Delta_0(x, a; \boldsymbol{p}) \approx \Delta_{0,\omega}(x, a; \boldsymbol{p}) = f(\psi(x))_a, \tag{3}$$
$$\Delta_i(x, a; \boldsymbol{p}) \approx \Delta_{i,\omega}(x, a; \boldsymbol{p}) = g(\psi(x), \phi(p_i), \phi(p_{i-1}))_a, \ i = 1, \cdots, N.$$

where $f : \mathbb{R}^d \to \mathbb{R}^{|\mathcal{A}|}$ and $g : \mathbb{R}^{2d} \to [0, \infty)^{|\mathcal{A}|}$ are two functions to be learned. $\omega$ includes all the parameters of the network.

Then, we use $\Pi_{\boldsymbol{p}, \boldsymbol{\Delta}_\omega}$ to denote a projection operator that projects the quantile function onto a piece-wise linear function supported by $\boldsymbol{p}$ and the incremental structure $\boldsymbol{\Delta}_\omega = \{\Delta_{0,\omega}, \cdots, \Delta_{N,\omega}\}$ above. For any quantile level $\tau \in (0,1)$, its projection-based quantile estimate is given by

$$F_{Z(x,a)}^{-1,\boldsymbol{p},\boldsymbol{\Delta}_\omega}(\tau) = \Pi_{\boldsymbol{p},\Delta_\omega} F_{Z(x,a)}^{-1}(\tau) = \Delta_{0,\omega}(x,a;\boldsymbol{p}) + \frac{1}{N}\sum_{i=0}^{N-1} G_{i,\boldsymbol{\Delta}_\omega}(\tau, x, a; \boldsymbol{p}), \quad (4)$$

where $G_{i,\boldsymbol{\Delta}_\omega}$ could be regarded as a linear combination of quantile estimates at two nearby supporting points satisfying

$$G_{i,\boldsymbol{\Delta}_\omega}(\tau, x, a; \boldsymbol{p}) = \left\{ \sum_{j=1}^{i} \Delta_{j,\omega}(x,a;\boldsymbol{p}) + \frac{\tau - p_i}{p_{i+1} - p_i}\Delta_{i+1,\omega}(x,a;\boldsymbol{p}) \right\} I(p_i \le \tau < p_{i+1}).$$

By limiting the output range of $g_a$ in (3) to be $[0,\infty)$, the obtained $N+1$ quantile estimates are non-decreasing. Under the NDQFN framework, the expected future return starting from $(x,a)$, also known as the $Q$-function can be empirically approximated by

$$Q_\omega = \int_{p_0}^{p_N} F_{Z(x,a)}^{-1,\boldsymbol{p},\boldsymbol{\Delta}_\omega}(\tau)d\tau = \sum_{i=0}^{N-1}(p_{i+1}-p_i)F_{Z(x,a)}^{-1,\boldsymbol{p},\boldsymbol{\Delta}_\omega}\left(\frac{p_i+p_{i+1}}{2}\right)$$

$$= \sum_{i=0}^{N-1} \frac{(p_{i+1}-p_i)}{2}\left[F_{Z(x,a)}^{-1,\boldsymbol{p},\boldsymbol{\Delta}_\omega}(p_{i+1}) + F_{Z(x,a)}^{-1,\boldsymbol{p},\boldsymbol{\Delta}_\omega}(p_i)\right]$$

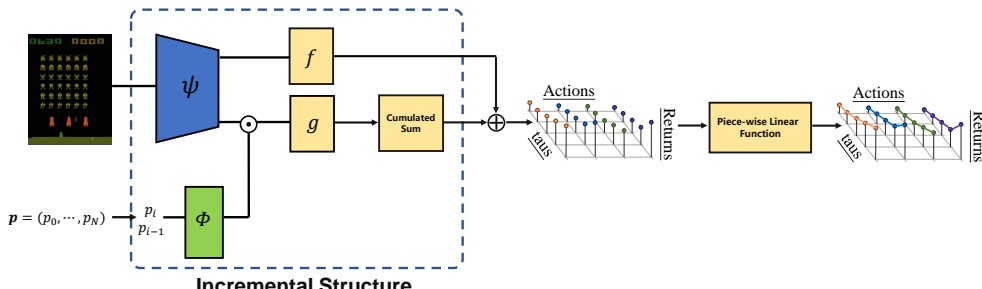

Figure 2: The network architecture of NDQFN.

For notational simplicity, we let $P_{\tau,\omega}(x,a) = F_{Z(x,a)}^{-1,\boldsymbol{p},\boldsymbol{\Delta}_\omega}(\tau)$, given the support $\boldsymbol{p}$. Following the idea of IQN, two random sets of quantile fractions $\boldsymbol{\tau} = \{\tau_1, \cdots, \tau_{N_1}\}$, $\boldsymbol{\tau}' = \{\tau_1', \cdots, \tau_{N_2}'\}$ are independently drawn from a uniform distribution $U(0,1)$ at each training iteration. In this case, for each $i \in \{1, \ldots, N_1\}$ and each $j \in \{1, \ldots, N_2\}$, the corresponding temporal difference (TD) error (Dabney et al., 2018a) with n-step updates on $(x_t, a_t, r_t, \cdots, r_{t+n-1}, x_{t+n})$ is computed as follows,

$$\delta_{i,j} = \sum_{i=0}^{n-1}\gamma^i r_{t+i} + \gamma^n P_{\tau_j',\omega'}\left(x_{t+n}, \arg\max_{a' \in \mathcal{A}} Q_{\omega'}(x_{t+n}, a')\right) - P_{\tau_i,\omega}(x_t, a_t), \quad (5)$$

where $\omega$ and $\omega'$ denote the online network and the target network, respectively. Thus, we can train the whole network by minimizing the Huber quantile regression loss (Huber, 1992) as follows,

$$\mathcal{L}(x_t, a_t, r_t, \cdots, r_{t+n-1}, x_{t+n}) = \frac{1}{N_2}\sum_{i=1}^{N_1}\sum_{j=1}^{N_2}\rho_{\tau_i}^\kappa(\delta_{i,j}), \quad (6)$$

where

$$\rho_\tau^\kappa(\delta_{i,j}) = |\tau - I(\delta_{i,j} < 0)|\frac{L_\kappa(\delta_{i,j})}{\kappa}, \quad (7)$$

$$L_\kappa(\delta_{i,j}) = \begin{cases} \frac{1}{2}\delta_{i,j}^2, & \text{if } |\delta_{i,j}| \le \kappa \\ \kappa\left(|\delta_{i,j}| - \frac{1}{2}\kappa\right), & \text{otherwise} \end{cases},$$

$\kappa$ is a pre-defined positive constant, and $|\cdot|$ denotes the absolute value of a scalar.

**Remark:** As shown by Figure 1 (c), the piece-wise linear structure ensures the monotonicity within the union of any two quantile sets $\boldsymbol{\tau}$ and $\boldsymbol{\tau}'$ from two different training iterations regardless of whether they are included in the $\boldsymbol{p}$ or not. However, as Figure 1 (b) demonstrates, directly applying a similar incremental structure onto IQN without using the piece-wise linear approximation may result in the non-monotonicity of $\boldsymbol{\tau} \cup \boldsymbol{\tau}'$ although each of their own monotonicity is not violated.

To investigate the convergence of the proposed algorithm, we introduce the following theorem, which can be seen an extension of Proposition 2 in Dabney et al. (2018b)

**Theorem 1.** *Let* $\Pi_{\boldsymbol{p},\boldsymbol{\Delta}_\omega}$ *be the quantile projection defined as above with non-decreasing quantile function* $F_Z^{-1,\boldsymbol{p},\boldsymbol{\Delta}_\omega}$. *For any two value distribution* $Z_1, Z_2 \in \mathcal{Z}$ *for an MDP with countable state and action spaces and enough large* $N$,

$$\bar{d}_\infty \left( \Pi_{\boldsymbol{p},\boldsymbol{\Delta}_\omega} \mathcal{T} F_{Z_1}^{-1,\boldsymbol{p},\boldsymbol{\Delta}_\omega}, \Pi_{\boldsymbol{p},\boldsymbol{\Delta}_\omega} \mathcal{T} F_{Z_2}^{-1,\boldsymbol{p},\boldsymbol{\Delta}_\omega} \right) \leq \gamma \bar{d}_\infty \left( F_{Z_1}^{-1,\boldsymbol{p},\boldsymbol{\Delta}_\omega}, F_{Z_2}^{-1,\boldsymbol{p},\boldsymbol{\Delta}_\omega} \right), \quad (8)$$

*where* $\mathcal{T}$ *denotes the distributional bellman operator,* $\bar{d}_k(F_{Z_1}^{-1}, F_{Z_2}^{-1}) := sup_{x,a} W_k(F_{Z_1(x,a)}^{-1}, F_{Z_2(x,a)}^{-1})$ *and* $W_k(\cdot,\cdot)$ *denotes the* $k$-*Wasserstein metric.*

By Theorem 1, we conclude that $\Pi_{\boldsymbol{p},\boldsymbol{\Delta}_\omega} \mathcal{T}$ have a unique fixed point and the repeated application of $\Pi_{\boldsymbol{p},\boldsymbol{\Delta}_\omega} \mathcal{T}$ converges to the fixed point. With $\bar{d}_k \leq \bar{d}_\infty$, the convergence occurs for all $k \in [1,\infty)$. It ensures that we can obtain a consistent estimator for $F_Z^{-1}$, denoted as $F_Z^{-1,\boldsymbol{p},\boldsymbol{\Delta}_\omega}$, by minimizing the Huber quantile regression loss defined in (6).

Now we discuss the implementation details of NQDFN. For the support $\boldsymbol{p}$ used in this work, we let $p_i = i/N$ for $i \in \{1, \cdots, N-1\}$, $p_0 = 0.001$ and $p_N = 0.999$. NDQFN models the state embedding $\psi(x)$ and the $\tau$-embedding $\phi(\tau)$ in the same way as IQN. The baseline function $f$ in (3) consists of two fully-connected layers and a sigmoid activation function, which takes $\psi(x)$ as the input and returns an unconstrained value for $\Delta_0(x,a;\boldsymbol{p})$. The incremental function $g$ shares the same structure with $f$ but uses the ReLU activation instead to ensure the non-negativity of all the $N$ increments $\Delta_i(x,a;\boldsymbol{p})$'s. Let $\odot$ denote the element-wise product and $g$ take $\psi(x) \odot \phi(p_i)$ and $\phi(p_i) - \phi(p_{i-1})$ as input, such that

$$\Delta_{i,\omega}(x,a;\boldsymbol{p}) = g(\psi(x) \odot \phi(p_i), \phi(p_i) - \phi(p_{i-1}))_a, \ i = 1, \cdots, N, \quad (9)$$

which to some extent captures the interactions among $\psi(x)$, $\phi(p_i)$ and $\phi(p_{i-1})$. Although $\psi(x) \odot \phi(p_i) \odot \phi(p_{i-1})$ may be another potential input form, the empirical results show that the combination of $\psi(x) \odot \phi(p_i)$ and $\phi(p_i) - \phi(p_{i-1})$ is more preferred in practice considering its outperformance in Atari games. More details are provided in Section C of the Supplement file.

## 5 EXPLORATION USING DISTRIBUTIONAL PREDICTION ERROR

To further improve the learning efficiency of NDQFN, we introduce a novel exploration approach called distributional prediction error (DPE), motivated by the Random Network Distillation (RND) method (Burda et al., 2018), which can be extended to most existing DRL frameworks. The proposed method, DPE, involves three networks, online network, target network and predictor network, which have the same architecture but different network parameters. The online network $\omega$ and the target network $\omega'$ use the same initialization, while the predictor network $\omega^*$ is separately initialized. To be more specific, the predictor network are trained on the sampled data using the quantile Huber loss defined as follows:

$$\mathcal{L}(x_t, a_t) = \frac{1}{N_2} \sum_{i=1}^{N_1} \sum_{j=1}^{N_2} \rho_{\tilde{\tau}_i}^\kappa \left( \delta_{i,j}^* \right),$$

where $\delta_{i,j}^* = P_{\tau_j^*,\omega'}(x_t, a_t) - P_{\tilde{\tau}_i,\omega^*}(x_t, a_t)$ and $\rho_\tau^\kappa(\cdot)$ are defined in (7). On the other hand, following Hasselt (2010) and Pathak et al. (2019), the target network of DPE is periodically synchronized by the online network. DPE employs the 1-Wasserstein metric $W_1(\cdot,\cdot)$ to measure the distance between the two quantile distributions associated with the predictor network and the target network. And its empirical approximation based on NDQFN is defined as follows,

$$i(x_t, a_t) = \int_{p_0}^{p_N} W_1 \left( P_{\tau,\omega'}(x_t, a_t) - P_{\tau,\omega^*}(x_t, a_t) \right) d\tau. \quad (10)$$

Since the minimum of $\mathcal{L}(x_t, a_t)$ is a contraction of the 1-Wasserstein distance (Bellemare et al., 2017), the prediction error would be higher for an unobserved state-action pair than for a frequently visited one. Therefore, $i(x_t, a_t)$ can be treated as the exploration bonus when selecting the optimal action, which encourages the agent to explore unknown states. With $i(x_t, a_t)$ being the exploration bonus, the optimal action is determined by

$$a_t = \arg\max_a [Q_\omega(x_t, a) + c_t i(x_t, a)], \tag{11}$$

where $c_t$ is the bonus rate. As might be expected, a more reasonable quantile estimate would highly enhance the exploration efficiency according to our exploration setting. To more clearly demonstrate this point, we compare the performance of DPE based on NDQFN and IQN in Section 6.2.

On the other hand, we compared the exploration efficiency of distributional prediction error and value-based prediction error in Section D of supplement, where value-based prediction error is defined as $i'(x_t, a_t) = |Q_{\omega*}(x_t, a_t) - Q_{\omega'}(x_t, a_t)|$. As might be expected, DPE enhances the exploration efficiency by utilizing more distributional information.

## 6 EXPERIMENTS

In this section, we evaluate the empirical performance of the proposed NDQFN with DPE on 55 Atari games using the Arcade Learning Environment (ALE) (Bellemare et al., 2013). The baseline algorithms we compare include IQN (Dabney et al., 2018a), QR-DQN (Dabney et al., 2018b), C51 (Bellemare et al., 2017), prioritized experience replay (Schaul et al., 2015) and Rainbow (Hessel et al., 2018). The whole architecture of our method is implemented by using the IQN baseline in the Dopamine framework (Castro et al., 2018). Note that the whole method could be built upon other quantile value based baselines such as FQF.

Our hyper-parameter setting is aligned with IQN for fair comparison. Furthermore, we incorporate n-step updates (Sutton, 1988), double Q learning (Hasselt, 2010), quantile regression (Koenker, 2005) and distributional Bellman update (Dabney et al., 2018b) into the training for all compared methods. The size of $p$ and the number of sampled quantile levels ,$\tau$ and $\tau'$, are 32, i.e. $N = N_1 = N_2 = 32$. We use $\epsilon$-greedy policy with $\epsilon = 0.01$ for training. At the evaluation stage, we test the agent for every 0.125 million frames with $\epsilon = 0.001$.

For the DPE exploration, we fix $c_t = 1$ without decay (which is the optimal setting based on experiment results). As a comparison, we extend DLTV (Mavrin et al., 2019), an exploration approach designed for DRL algorithms based on fixed quantile locations such as QR-DQN, to quantile value based methods by modifying the original left truncated variance used in their paper to $\sigma_+^2 = \int_{\frac{1}{2}}^1 [F_Z^{-1}(\tau) - F_Z^{-1}(\frac{1}{2})]^2 d\tau$. All training curves in this paper are smoothed with a moving average of 10 to improve the readability. With the same hyper-parameter setting, NDQFN with DPE is about 25% slower than IQN at training stage due to the added predictor network.

### 6.1 NDQFN VS IQN

In this part, we compare NDQFN with its baseline IQN (Dabney et al., 2018a) to verify the improvement it achieves by employing the non-decreasing structure. The two methods are fairly compared without employing any additional exploration strategy.

Figure 3 visualizes the training curves of eleven randomly selected easy explored Atari games. NDQFN significantly outperforms IQN in most cases, including Berzerk, KungFu Master, Ms. Pac-Man, Qbert, River Raid, Up and Down, Video Pinball and Zaxxon. Although NDQFN achieves similar final scores with IQN in Battle Zone and Tennis, it learns much faster. This tells that the crossing issue, which usually happens in the early training stage with limited training samples, can be sufficiently addressed by the incremental design in NDQFN to ensure the monoticity of the quantile estimates. More related results could be seen in Section E of the supplement.

### 6.2 EFFECTS OF NDQFN ON DPE

In this section, we show how the NDQFN design can help improve the exploration efficiency of DPE compared to using the IQN baseline. We pick three hard explored games, Montezuma Revenge,

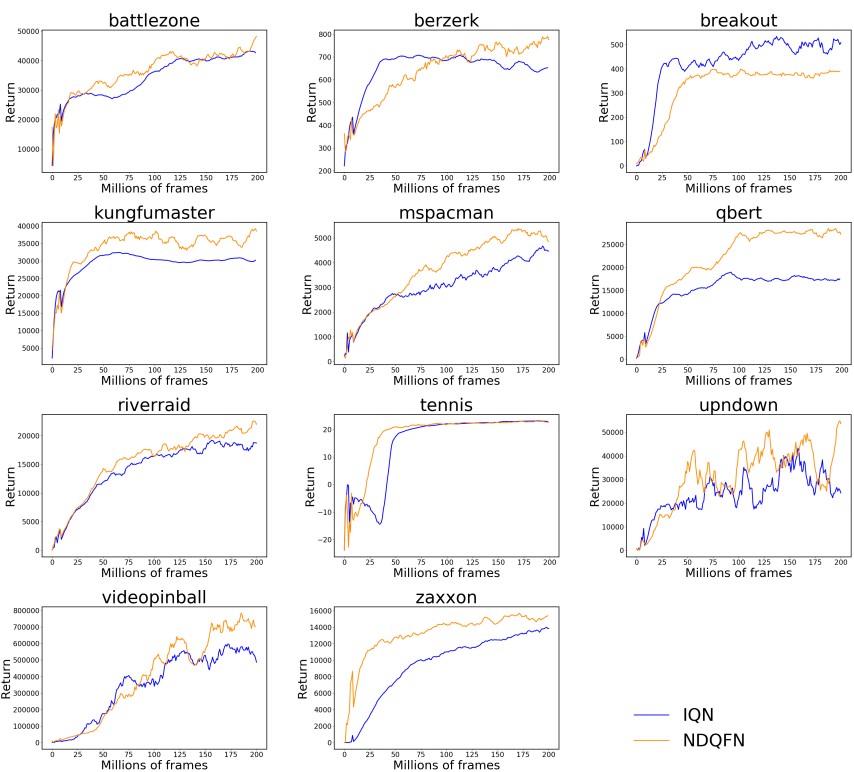

Figure 3: Training curve on Atari games for NDQFN and IQN.

Private Eye, Venture, and three easy explored games, Ms. Pac-Man, River Raid and Up and Down. The main results are summarized in Figure 4.

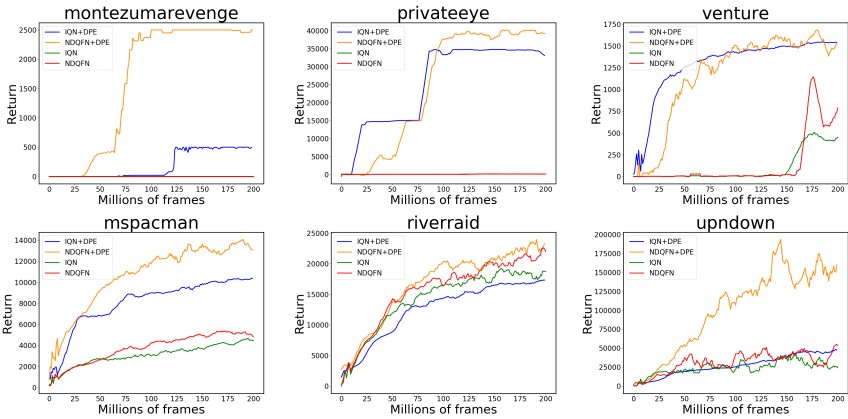

Figure 4: Comparison between NDQFN+DPE and IQN+DPE.

As Figure 4 illustrates, DPE is an effective exploration method which highly increases the training performance of both IQN and NDQFN in three hard explored games. In the three easy-explored games, DPE still performs slightly better than the baseline methods, which to some extent demonstrates the stability and robustness of DPE. On the other hand, we find that NDQFN + DPE significantly outperforms IQN + DPE especially in some hard-explored games. This agrees with our conclusion in Section 5 that NDQFN obtains a more reasonable quantile estimate by adding the non-decreasing constraint which helps to increase the exploration efficiency.

## 6.3 DPE VS DLTV

We do some more experiments to show the advantage of DPE over DLTV when applying both of the two exploration approaches to quantile value based DRL algorithms. To be specific, we evaluate the performance of NDQFN+DPE and NDQFN+DLTV on the six games examined in the previous subsection. Figure 5 shows that DPE achieves a much better performance than DLTV especially in the three hard explored games. DLTV does not achieve significant performance gain over the baseline NDQFN without doing exploration.

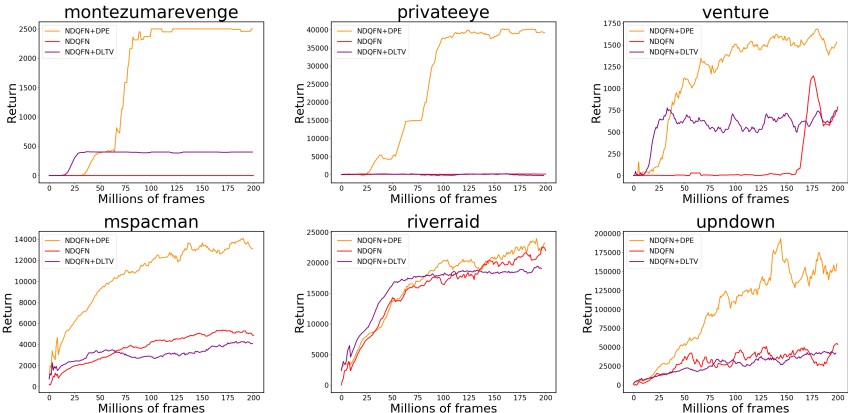

Figure 5: Comparison between DPE and DLTV.

This result tells that DLTV, which computes the exploration bonus based on some discrete quantile locations, performs extremely unstable for quantile value based methods such as IQN and NDQFN since the quantile locations used to train the model are changed each time. As a contrast, DPE focuses on the entire distribution of the quantile function, which relies less on the selection of the quantile fractions, and thus performs better in this case.

## 6.4 Full Atari Results

In this part, we evaluate the performance of the whole network which incorporates NDQFN with DPE in all the 55 Atari games. Following the IQN setting (Dabney et al., 2018a), all evaluations start with 30 non-optimal actions to align with previous distributional RL works. 'IQN' in the table tries to reproduce the results presented in the original IQN paper, while 'IQN*' corresponds to the improved version of IQN by using n-step updates and some other tricks mentioned above.

|  | Mean | Median | >Human |
|---|---|---|---|
| DQN | 221% | 79% | 24 |
| PRIOR. | 580% | 124% | 39 |
| C51 | 701% | 178% | 40 |
| RAINBOW | 1213% | 227% | 42 |
| QR-DQN | 902% | 193% | 41 |
| IQN | 1112% | 218% | 39 |
| IQN* | 1207% | 224% | 40 |
| NDQFN+DPE | **1959%** | **237%** | **42** |

Table 1: Mean and median of scores across 55 Atari 2600 games, measured as percentages of human baseline. Scores of previous work are referenced from Castro et al. (2018) and Yang et al. (2019).

Table 1 presents the mean and median human normalized scores by the seven compared methods across 55 Atari games, which shows that NDQFN with DPE significantly outperforms the IQN baseline and other state-of-the-art algorithms such as Rainbow. In particular, Figure 6 plots the

relative improvement of NDQFN+DPE over IQN ((NDQFN-random)/(IQN-random)-1) in testing score for all the 55 Atari games. We observe that our method consistently outperforms the IQN baseline in most situations, especially for some hard games with extremely sparse reward spaces, such as Private Eye and Montezuma Revenge. More comparison results including the training curves for all the 55 games and the raw scores are provided in Sections F and G in the supplement.

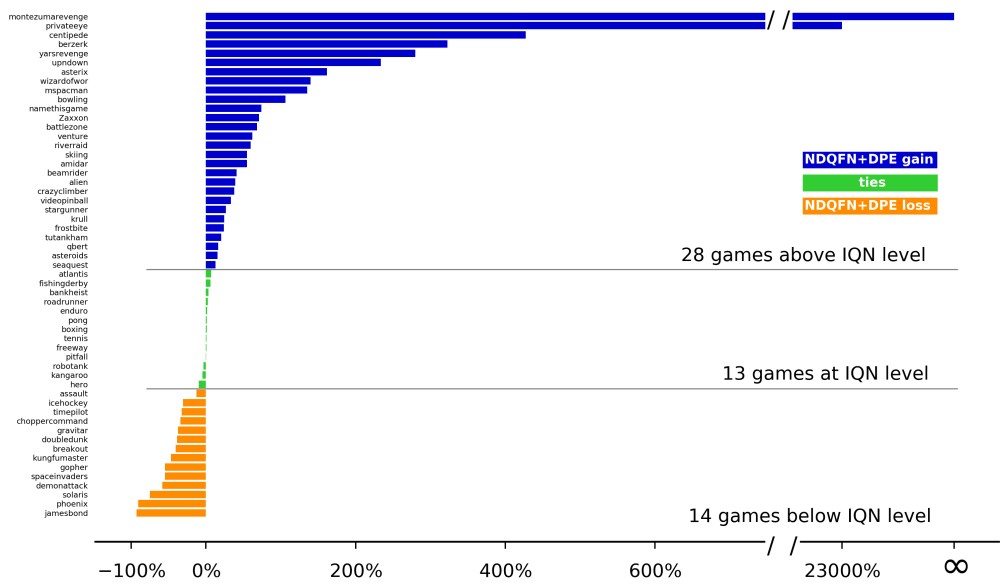

Figure 6: Cumulative rewards performance comparison between our method and IQN with 200M training frames. The bars represent relative gain/loss of NDQFN+DPE over IQN.

## 7 CONCLUSION AND DISSCUSION

In this paper, we make some attempts to address two important issues people care about for distributional reinforcement learning studies. We first propose a Non-decreasing Quantile Function Network (NDQFN) to ensure the validity of the learned quantile function by adding the monotonicity constraint to the quantile estimates at different locations via a piece-wise linear incremental structure. We also introduce DPE, a general exploration method for all kinds of distributional RL frameworks, which utilizes the information of the entire distribution and performs much more efficiently than the existing methods. Experiment results on more than 50+ Atari games illustrates NDQFN can more precisely model the quantile distribution than the baseline methods. On the other hand, DPE perform better in both hard-explored Atari games and easy ones than DLTV. Some ablation studies show that the combination of NDQFN with DPE achieves the best performance than all the others.

There are some questions remain unsolved. First, since lots of non-negative functions are available, is Relu a good choice for function $g$? Second, will a better approximated distribution affect agent's policy? If so, how does it affect the training process as NDQFN can provide a potentially better distribution approximation. More generally, how to analytically compare the optimal policies NDQFN and IQN converge to. Third, how much does the periodically synchronized target network affect the efficiency of the DPE exploration.

For the future work, we first want to apply NDQFN to other quantile value based methods such as FQF to see if the incremental structure can help improve the performance of all the existing DRL algorithms. Second, as there exists some other kind of DRL methods which naturally ensures the monotonicity of the learned quantile function (Yue et al., 2020) but requies extremely large computation cost, it may be interesting to integrate our method with these methods to obtain an improved empirical performance and training efficiency.

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
