# OpenReview forum: "Non-decreasing Quantile Function Network with Efficient Exploration for Distributional Reinforcement Learning"
_ICLR.cc/2021/Conference — Reject_

### Official Review · AnonReviewer3 · 2020-10-20

**Rating:** 6
**Confidence:** 5

**Review:**

Paper Summary: This paper mainly contributes in two parts: 1). A non-decreasing structure for quantile values in quantile-based distributional RL. 2). A curiosity-based intrinsic reward using distribution disagreement.

Clarity:
- Some mathematical expressions, while correct, are hard to interpret, e.g. G_{i,\delta}, H_{i,\omega', \omega^*}. It would be nice if the authors could provide a brief explanation in words. For example, G_{i,\delta} is just linear interpolation given fixed support.

- The paper stops at experiment results. There are no further discussions or conclusion section.

- The relation between RND and proposed DPE is not as close as the authors suggest. In RND, the target network is a fixed network for the predictor network to mimic. In DPE, both the target network and the predictor is trained with the same objective, i.e. minimize TD error. The intrinsic reward is evaluated by the disagreement between the predictor and the target network. Such disagreement-based intrinsic reward had been studied in other lines of work, e.g. [1].

Pros:
- Resolves existing crossing quantile issue in most quantile-based distributional RL algorithms.
- Proposed method ensures monotonicity on all possible quantile sets. Using linear interpolation for distribution approximation feels natural and straightforward.
- Distribution based curiosity is novel.

Cons:
- The significance of DPE is not properly evaluated. In fact, the paper only provide detailed results for NDFQN+DPE. It is hard to tell exactly how much of the performance gain is credit to NDFQN or DPE. To evaluate the significance of DPE, I would expect the authors to compare across different curiosity-driven exploration methods on the same baseline, say IQN. At least, the authors should answer how much distribution-based disagreement outperforms value based disagreement.

- Same reason as above, the performance gain of NDFQN itself cannot be inferred from just 6 games. Judging from section 4.1 and 4.2, I would presume that the performance gain is mainly credit to DPE instead of NDFQN.

- The issue in figure 1(c) is not entirely resolved. Different from the issue in figure 1(b), this issue directly comes from using incrementals instead of values. The authors partially resolves it by using a fixed set of support, but if the support is changeable the issue remains.


Questions:
- Is the embedding of p* really necessary? Or, is p* necessary for network input? As the support p* is fixed, I do not believe that they should be part of the input. From my own point of view, using modified QR-DQN's structure sounds more reasonable. If the authors find out that an additional, fixed input does impact the performance significantly, some explanations or insights are required.

Post rebuttal:
- The authors have addressed most of my main concerns and the additional experiment looks good to me. Therefore, I increase my score to 6. However, some experiment results are still missing and the paper still needs some editing before published, especially the experiment section. For example figure 6. is misleading since the comparison is not fair.

References:
[1] Pathak, Deepak, Dhiraj Gandhi, and Abhinav Gupta. "Self-Supervised Exploration via Disagreement." International Conference on Machine Learning. 2019.

---

> ### Author Response · Authors · 2020-11-25
> **Responses to the Comments of Reviewer 4 (Part 1)**
>
> Thank you for the thoughtful and constructive suggestions. We have taken all the comments into consideration and summarize the responses in the following. Belows are our point to point responses to your comments.
>
> **1.** Some mathematical expressions, while correct, are hard to interpret, e.g. G_{i,\delta}, H_{i,\omega', \omega^*}. It would be nice if the authors could provide a brief explanation in words. For example, G_{i,\delta} is just linear interpolation given fixed support.
> The paper stops at experiment results. There are no further discussions or conclusion section.
>
> **Response**: Thank you for your good suggestion. We have revised the paper carefully to address these questions. In the revised manuscript, we add more explanations to interpret some important mathematical expressions, such as Line 3 on Page 4 (marked as red). Besides, in Section 7 we summarize the major contributions in this paper and discuss some unsolved problems and possible future work.
>
> **2**.The relation between RND and proposed DPE is not as close as the authors suggest. In RND, the target network is a fixed network for the predictor network to mimic. In DPE, both the target network and the predictor are trained with the same objective, i.e. minimize TD error. The intrinsic reward is evaluated by the disagreement between the predictor and the target network. Such disagreement-based intrinsic reward had been studied in other lines of work, e.g. [1].
>
> **Response**: Thank you for pointing out this problem. We choose a periodically synchronized target network in DPE, since it is required by implementing double Q-learning and performs well in practice. We need to thank the anonymous reviewer, the idea of Pathak et. al (2019) is indeed more related to ours and we have already cited this paper in our revised manuscript.
>
> **3.** The significance of DPE is not properly evaluated. In fact, the paper only provides detailed results for NDFQN+DPE. It is hard to tell exactly how much of the performance gain is credit to NDFQN or DPE. To evaluate the significance of DPE, I would expect the authors to compare across different curiosity-driven exploration methods on the same baseline, say IQN. At least, the authors should answer how much distribution-based disagreement outperforms value-based disagreement.
>
> **Response**: Thank you for your good suggestion. Following your suggestion, we compare the value-based disagreement with distribution-based disagreement in Appendix. The exploration bonus of value-based disagreement is defined as follow:
> $$i'(x_t,a_t) = |Q_{\omega^*}(x_t,a_t) - Q_{\omega'}(x_t,a_t)|,$$
> where $Q_{\omega'}$ and $Q_{\omega^*}$ denote the state-action value function of the target and predictor network, respectively. As Figure 3 in the supplement shows, the distribution-based disagreement significantly outperforms value-based disagreement on three hard-explored games with 100M training frames.
>
> **4.** Same reason as above, the performance gain of NDFQN itself cannot be inferred from just 6 games. Judging from section 4.1 and 4.2, I would presume that the performance gain is mainly credit to DPE instead of NDFQN.
>
> **Response**: Thank you for your comment. Due to the limited time, we cannot provide the full 200M results of all the 55 Atari game. In Figure 3 of Section 6.1 and Figure 2 of the appendix, we compare 29 games while some are based on 200M and some are based on 100M. According to the existing results, we can see that the non-decreasing constraint helps NDQFN to achieve better performance than the baseline IQN in most cases among 29 Atari games. We will definitely add the full the results into the final version if the paper is accepted.

---

> ### Author Response · Authors · 2020-11-25
> **Responses to the Comments of Reviewer 4 (Part 2)**
>
> **5.** The issue in figure 1(c) is not entirely resolved. Different from the issue in figure 1(b), this issue directly comes from using incrementals instead of values. The authors partially resolves it by using a fixed set of support, but if the support is changeable the issue remains.
>
> **Response**: Thank you for your comment. Exactly, we consider p as a hyper-parameter of NDQFN like QR-DQN (Dabney et al., 2018a) and we show how we choose the supporting set in both the end of Section 4 and the start of Section 6.
>
> **6.** Is the embedding of p* really necessary? Or, is p* necessary for network input? As the support p* is fixed, I do not believe that they should be part of the input. From my own point of view, using modified QR-DQN's structure sounds more reasonable. If the authors find out that an additional, fixed input does impact the performance significantly, some explanations or insights are required.
>
> **Response**: Thank you for your comment. The results in Dabney et al. (2018b) shows that IQN achieves significant improvement on QRDQN. Motivated by this, we consider the similar network with IQN and similar training process, which contains an embedding of quantile fractions. NDQFN can actually be seen as the improvement of the combination of QRDQN and IQN. The supporting points are fixed like QRDQN, while the quantile fractions used for model training are resampled each time to make the network more closely approximate the entire distribution of the target quantile function. With the help of these varied τ from different training iterations, the final quantile estimates at the support points p* may be more precise than QRDQN.
>
> [1] Will Dabney, Mark Rowland, Marc G Bellemare, and R ́emi Munos. Distributional reinforcement learning with quantile regression. In AAAI, 2018a.
>
> [2] Will Dabney, Georg Ostrovski, David Silver, and Remi Munos. Implicit quantile networks for distributional reinforcement learning. In International Conference on Machine Learning, pp.1096–1105, 2018b.
>
> [3] Deepak Pathak, Dhiraj Gandhi, and Abhinav Gupta. Self-supervised exploration via disagreement. In International Conference on Machine Learning, pp. 5062-5071, 2019.

---

### Official Review · AnonReviewer2 · 2020-10-27
**Interesting idea but not yet sufficiently proper experiments to understand the role of NDQFN and DPE separately, and fairer comparison with the baseline IQN is required**

**Rating:** 5
**Confidence:** 4

**Review:**

###  Summary
This paper proposes to use a monotonic quantile function for distributional RL by: (i) estimating the quantile values at the supported quantiles via a cumulative sum of non-negative incremental value, and (ii) interpolate the quantile values at the unsupported quantiles via linear combination of the nearest supported quantiles. The paper then also proposes to estimate the exploration bonus for each state using random network distillation. It then conducts experiments in Atari games to verify some conclusions.

### Strong points
-	Clarity: The paper is well structured.
-	Technical novelty: immediate. The idea of piece-wise linear function to approximate quantile function in Eq. (4) seems novel though very natural. The idea of using random network distillation to measure the exploration bonus seems novel and useful.
-	Empirical significance: The combination of DPE with NDQFN performing well in some hard-explored games (Fig. 3) looks very encouraging and promising. In my opinion, I think this is the most interesting and significant empirical result reported in this paper in the current form.

###  Weak points
-	The experimental comparison/conclusion with IQN in the present form is unfair.
In particular, after the Table 1, the authors conclude that “NDQFN with DPE significantly outperforms the IQN baseline”. I think this is an unfair conclusion as the result of IQN reported in Table 1 is merely the performance for distributional part, i.e., the IQN result in Table 1 does not include any orthogonal improvements such as n-updates, double networks and any advanced exploration method beyond epsilon-greedy while NDQFN incorporates DPE exploration method (and n-step updates  and double Q update if I infer correctly from the paper). To be fair, I think the paper should include to IQN any orthogonal improvements that NDQFN has.  Even though Figure 2 has a fair comparison between NDQFN and IQN where both use n-step updates and without DPE exploration, I think only 6 games are not sufficient to make a reliable conclusion that the proposed non-decreasing quantile function is more helpful than the original quantile function in IQN.
-	It is unclear from the experimental results how helpful are each non-decreasing quantile function and DPE individually.
Except a nice result in Figure 3, I am not fully convinced if the proposed non-decreasing quantile function is actually more helpful than the original quantile function as a distributional component. For example, I think it is more useful to report the result of a mere NDQFN (i.e., without DPE or any exploration methods rather than epsilon-greedy, without any orthogonal improvements such as n-step updates and double Q value) in the full Atari games to see how much it improves over the original IQN (reported in Figure 1). Regarding DPE, since it is a new exploration method, I think it is more helpful and reliable to experimentally compare DPE with some other distributional exploration methods such as  DLTV (Mavrin et al. 2019). I think the current experimental results are hard to make any reliable conclusion about each NDQFN and DPE individually except a nice observation that a combination of NDQFN and DPE can help in some hard explored games.

###  Questions for the authors.
-	Though I believe that IQN indeed does not guarantee the monotonicity in quantile values it estimates, the illustrative figure for this (Fig 1) seems a hypothetical one instead of a real experimental result.  Can we show that empirically by a simulation?
-	Does the NDQFN+DPE in Table 1 also have n-step update and double Q network update style?

###  My initial recommendation
For the main reasons in the weak points section, I vote for rejecting for this current form.

### My final recommendation

The authors have attempted to address some of my points but these points require more time to fully address as they require to run more experiments. For the current form, I remain my inital score and recommend rejection for this time.

---

> ### Author Response · Authors · 2020-11-25
> **Responses to the Comments of Reviewer 3 (Part 1)**
>
> Thank you for the thoughtful and constructive suggestions. We have taken all the comments into consideration and summarize the responses in the following. Below are our point to point responses to your comments.
>
> **1.** The experimental comparison/conclusion with IQN in the present form is unfair. In particular, after the Table 1, the authors conclude that “NDQFN with DPE significantly outperforms the IQN baseline”. I think this is an unfair conclusion as the result of IQN reported in Table 1 is merely the performance for distributional part, i.e., the IQN result in Table 1 does not include any orthogonal improvements such as n-updates, double networks and any advanced exploration method beyond epsilon-greedy while NDQFN incorporates DPE exploration method (and n-step updates and double Q update if I infer correctly from the paper). To be fair, I think the paper should include to IQN any orthogonal improvements that NDQFN has. Even though Figure 2 has a fair comparison between NDQFN and IQN where both use n-step updates and without DPE exploration, I think only 6 games are not sufficient to make a reliable conclusion that the proposed non-decreasing quantile function is more helpful than the original quantile function in IQN.
>
> **Response**: Thank you for your comment. In the revised manuscript, we provide the results of IQN with n-step updates and double Q-learning, labeled as IQN*, in Table 1 for a fair comparison.
> Due to the limited time, we cannot provide the full comparison between NDQFN and IQN on all the 55 Atari games. However, we provide 200 M results for 11 games in Figure 3 and 100 M results for the other 18 games in Figure 2 of the appendix. The remaining part of experimental results would be done in the future. According to the existing results, we can see that the non-decreasing constraint helps NDQFN to achieve better performance than the baseline IQN in most cases among 29 Atari games. Moreover, as Figure 4 in Section 6.2 shows, NDQFN works better with the proposed DPE exploration than IQN since NDQFN obtains a more reasonable quantile estimate by adding the non-decreasing constraint which helps to increase the exploration efficiency.
>
> **2.** It is unclear from the experimental results how helpful are each non-decreasing quantile function and DPE individually. Except a nice result in Figure 3, I am not fully convinced if the proposed non-decreasing quantile function is actually more helpful than the original quantile function as a distributional component. For example, I think it is more useful to report the result of a mere NDQFN (i.e., without DPE or any exploration methods rather than epsilon-greedy, without any orthogonal improvements such as n-step updates and double Q value) in the full Atari games to see how much it improves over the original IQN (reported in Figure 1). Regarding DPE, since it is a new exploration method, I think it is more helpful and reliable to experimentally compare DPE with some other distributional exploration methods such as DLTV (Mavrin et al. 2019). I think the current experimental results are hard to make any reliable conclusion about each NDQFN and DPE individually except a nice observation that a combination of NDQFN and DPE can help in some hard-explored games.
>
> **Response**: Thank you very much for this good suggestion. In the revised version, we evaluate the empirical performance of NDQFN with DLTV and compare it with NDQFN +DPE. We extend DLTV (Mavrin et al., 2019), an exploration approach designed for DRL algorithms using fixed quantile locations such as QR-DQN, to quantile value based methods by modifying the original left truncated variance to
> $$\sigma_+^2 = \int_{\frac{1}{2}}^1 [F_Z^{-1}(\tau) - F_Z^{-1}(\frac{1}{2})]^2 d\tau.$$
> Figure 5 shows that DLTV method perform well in early stage with limited training samples. However, DPE achieves a much better performance than DLTV especially in the three hard explored games. This tells that DPE more sufficiently utilize the entire distribution of the quantile function than DLTV.
>
> **3.** Though I believe that IQN indeed does not guarantee the monotonicity in quantile values it estimates, the illustrative figure for this (Fig 1) seems a hypothetical one instead of a real experimental result. Can we show that empirically by a simulation?
>
> **Reponse**: Thank you for your comment. In Figure 1, we present a real quantile curve obtained by $N$ uniformly sample quantile fractions at a certain $(s,a)$ pair estimated by IQN under 50M training on Breakout. As the figure shows, the quantile curve is obviously non- monotonic. As a comparison, our method can ensure the monotinicity of the quantile estimates regardless of the selection of $\tau$.

---

> ### Author Response · Authors · 2020-11-25
> **Responses to the Comments of Reviewer 3 (Part 2)**
>
> **4.**	Does the NDQFN+DPE in Table 1 also have n-step update and double Q network update style?
>
> **Reponse**: Thank you for your comment. Yes, we emphasize in the second paragraph of Section 6 (marked by red) that we incorporate n-step updates (Sutton, 1988), double Q-learning (Hasselt, 2010), quantile regression (Koenker,2005) and distributional Bellman update (Dabney et al., 2018a) into the training for all compared methods
>
> [1] Will Dabney, Mark Rowland, Marc G Bellemare, and R ́emi Munos. Distributional reinforcement learning with quantile regression. In AAAI, 2018a.
>
> [2] Will Dabney, Georg Ostrovski, David Silver, and Remi Munos. Implicit quantile networks for distributional reinforcement learning. In International Conference on Machine Learning, pp.1096–1105, 2018b.
>
> [3] Borislav Mavrin, Shangtong Zhang, Hengshuai Yao, Linglong Kong, Kaiwen Wu, and Yaoliang Yu. Distributional reinforcement learning for efficient exploration. In International Conference on Machine Learning, pp. 4424–4434, 2019.
>
> [4] Richard S Sutton. Learning to predict by the methods of temporal differences. Machine learning, 3(1):9–44, 1988.
>
> [5] Hado V Hasselt. Double q-learning. In Advances in neural information processing systems, pp.2613–2621, 2010.
>
> [6] Koenker. Quantile regression. Cambridge University Press, 2005.

---

### Official Review · AnonReviewer4 · 2020-10-28

**Rating:** 4
**Confidence:** 4

**Review:**

Summary:

The paper provides a method to learn the quantile function in distributional reinforcement learning that guarantee its monotonicity. Moreover, an exploration strategy for distributional reinforcement learning is also presented.

Reasons for score:

The paper claims two contributions: a better way to model quantile functions and an exploration strategy for distributional reinforcement learning. The former is a trivial extension of previous work, and the latter is not properly explained.

Pros:

1. The experiment results look good.

2. The DPE exploration strategy seems to be effective in the examples.


Cons:

1. The NDQFN is a trivial extension of (3) through piece-wise linear interpolation. The explanation of Figure 1 is misleading.

2. There is another class of methods in distributional reinforcement learning which uses generative network to model the Z function, which doesn’t suffer from lack of monotonicity as in quantile function learning (see e.g. Y. Yue, Z. Wang, M. Zhou ``Implicit Distributional Reinforcement Learning”). These type of methods need to be properly discussed and also compared in the experiments.

3. Theorem 1 could be problematic. The quantile projection defined in this paper doesn’t seem to be contractive.

4. Section 3.4 is not well presented.

Questions:

Please address and clarify the cons above.

---

> ### Author Response · Authors · 2020-11-25
> **Responses to the Comments of Reviewer 2**
>
> Thank you for the thoughtful and constructive suggestions. We have taken all the comments into consideration and summarize the responses in the following. Below are our point to point responses to your comments.
>
> **1.** The NDQFN is a trivial extension of (3) through piece-wise linear interpolation. The explanation of Figure 1 is misleading.
>
> **Response**: Thank you for your comment. We may not make the statements clear in this part. We rewrite the introduction part of NDQFN in Section 4. The incremental structure in (3) is actually an important component of NDQFN architecture, which is proposed by this paper. However, it cannot completely solve the crossing issue by itself without the piece-wise linear structure. Also, we modify Figure 1 to make it more interpretable and add more detailed explanation about it in a Remark before Theorem 1. directly applying a similar incremental structure onto IQN without using the piece-wise linear approximation may result in the non-monotonicity of τ∪τ^' although each of their own monotonicity is not violated.
>
> **2.** There is another class of methods in distributional reinforcement learning which uses generative network to model the Z function, which doesn’t suffer from lack of monotonicity as in quantile function learning (see e.g. Y. Yue, Z. Wang, M. Zhou ``Implicit Distributional Reinforcement Learning”). These type of methods need to be properly discussed and also compared in the experiments.
>
> **Response**: Thank you for pointing out this work. Yue et al. (2020) adapts a distributional perspective on the discounted cumulative return and model it with a state-action-dependent implicit distribution, which is approximated by the DGNs that take state-action pairs and random noises as their input. However, the proposed method would make more time consumption than NDQFN, since the DGNs inputs $K$ random noises and sorting the $K$ elements of output to compute empirical distribution. The sorting process could be time wasted. Since this new method does provide results on Atari and the pre-released code on Github seems to be problematic, we do not include its empirical comparison in the experiment section due to the limited time. However, we still add the citation of this paper into the revised paper and consider dosing come comparison in the future if possible.
>
> **3.** Theorem 1 could be problematic. The quantile projection defined in this paper doesn’t seem to be contractive.
>
> **Response**: Thank you for pointing out this problem. Exactly, under some conditions on the dimension of support p and sample number of τ, Theorem 1 could be seen an extension of Proposition 2 in Dabney et al., (2018a). The main idea of the proof is that if N is large enough, each element of sampled τ would fall into one of the N [p_i,p_(i+1)]’s, and the proof would be similar to the proof of Proposition 2 in Dabney et al., (2018a). The exhaustive proof would be provided in the final version if the paper is accepted.
>
> **4.** Section 3.4 is not well presented.
>
> **Response**: Thank you for pointing out this problem. We have rewritten this section in the revised manuscript. In Section 5, we first describe the architecture of DPE, which includes three networks. Then, we describe the training process of the predictor network and provide an empirical approximation used by DPE. Finally, we present how the DPE design can help improve the exploration efficiency compared to using the value-based prediction error.
>
> [1] Will Dabney, Mark Rowland, Marc G Bellemare, and R ́emi Munos. Distributional reinforcement learning with quantile regression. In AAAI, 2018a.
>
> [2] Yuguang Yue, Zhendong Wang, and Mingyuan Zhou. Implicit distributional reinforcement learning. Advances in Neural Information Processing Systems, 33, 2020.

---

### Official Review · AnonReviewer1 · 2020-10-30
**Interesting alg that extends DLTV to IQN**

**Rating:** 6
**Confidence:** 5

**Review:**

This paper studies distributional RL and proposed two extensions. One is a method to enforce a non-decreasing ordering of quantile functions by a linear and non-negative increments. The other is extends the idea of DLTV which adds exploration bonus in action selection by using the random network distillation method, which in particular, using a measure of inconsistency between target network and predictor networks as a frequency measure of sampled states.

How do you convince us enforcing a non-decreasing ordering of the learned quantile functions is helpful?
I understand your arguments, but there is no evidence in the paper showing that doing so is helpful.

Comparison with DLTV is missing.
The paper argues that DLTV is not applicable to continuous quantiles. However, it would be to include this comparison especially they have results on Atari games as well.

The empirical results are not very strong, with 13 and 14 ties and losses with/to IQN. It appears the advantage of DLTV over QRDQN is larger than your advantage over IQN.

The technical quality and presentation of the paper can still be much improved.

Abstract:
two important problems still remain unsolved.
the other is how to design
an efficient exploration strategy to fully utilize the distribution information
--> Later you showed this is false argument by introducing DLTV (Mavrin et. al. 19)

We describe the implementation details of the two architectures with
what are they?
you have two "architectures"?

 (b)(c) a simple incremental structure proposed in(3):
this sentence is confusing.

What is the circle dot operator? (.)

Eq 4 is just interpolation to ensure positive increments.
Why do you need to show (3) since it's not good?

Th 1:
The definition of the \Pi operator isn't clear.

Before Sec 3.4:

Is Relu is a good choice? What is your thought on other functions? How to choose g in practice?


The prediction error would be high
->The prediction error would be higher

eq 11:
This is similar to Mavrin's idea: using exploration bonus -- UCT style.

---

> ### Author Response · Authors · 2020-11-25
> **Responses to the Comments of Reviewer 1**
>
> Thank you for the thoughtful and constructive suggestions. We have taken all the comments into consideration and summarize the responses in the following. Below are our point to point responses to your comments.
>
> **1.**	How do you convince us enforcing a non-decreasing ordering of the learned quantile functions is helpful? I understand your arguments, but there is no evidence in the paper showing that doing so is helpful.
>
> **Response**: Thank you for your comment. We add more detailed comparison between NDQFN and IQN in the experiment part. Figure 3 in Section 6.1 and Figure 2 in the supplement show that the non-decreasing constraint helps to achieve better performance in most cases among 29 Atari games. As Figure 4 in Section 6.2 shows, NDQFN works better with the proposed DPE exploration than IQN since NDQFN obtains a more reasonable quantile estimate by adding the non-decreasing constraint which helps to increase the exploration efficiency. The complete results for all the 55 games will be included if the paper is accepted.
>
> **2.**	Comparison with DLTV is missing. The paper argues that DLTV is not applicable to continuous quantiles. However, it would be to include this comparison especially they have results on Atari games as well.
> The empirical results are not very strong, with 13 and 14 ties and losses with/to IQN. It appears the advantage of DLTV over QRDQN is larger than your advantage over IQN.
>
> **Response**: Thank you very much for this good suggestion. We add the comparison between DLTV and DPE with NDQFN being the baseline. Since DLTV is originally designed for DRL algorithms based on fixed quantile locations such as QR-DQN, we modify the function form of the left truncated variance to work for quantile value-based methods such as IQN where the quantile fractions being sampled at each training iteration are different:
> $$\sigma_+^2 = \int_{\frac{1}{2}}^1 [F_Z^{-1}(\tau) - F_Z^{-1}(\frac{1}{2})]^2 d\tau$$
> The Figure 5 shows that DLTV method perform well in early stage with limited training samples. However, DPE achieves a much better performance than DLTV especially in the three hard explored games.
> Also, we want to emphasize that the Figure 10 in Mavrin et al., (2019) is evaluated on 40 million frames while ours based on 200 million frames. Since the training process on the early stage is unstable, we think it is more reasonable to draw conclusion based on 200 million training frames, like Figure 6 in our paper.
>
> **3.**	The technical quality and presentation of the paper can still be much improved. Abstract: two important problems still remain unsolved. the other is how to design an efficient exploration strategy to fully utilize the distribution information --> Later you showed this is false argument by introducing DLTV (Mavrin et. al. 19)
> We describe the implementation details of the two architectures with what are they? you have two "architectures"?
> (b)(c) a simple incremental structure proposed in(3): this sentence is confusing.
> What is the circle dot operator? (.)
>
> **Response**: Thank you for your comment. We have revised the paper carefully to address these questions. Please see Abstract, Line 7 on Page 2, modified Figure 1 on Page 3, Remark on Page 5 and Line 25 on Page 5 in the updated paper, which are marked by the red color for details.
>
> **4.** Eq 4 is just interpolation to ensure positive increments. Why do you need to show (3) since it's not good?
>
> **Response**: We thank you for pointing out this problem. The original writing of this part may be a little confusing. The incremental structure in (3) is actually an important component of the proposed NDQFN architecture, while it cannot completely solve the crossing issue by itself (shown by the modified Figure 1). We revise Section 4 to make it more interpretable now and add a Remark before Theorem 1 to explain why the piece-wise linear structure is necessary.
>
> **5.**	Th 1: The definition of the \Pi operator isn't clear.
>
> **Response**: Thank you for your comment. $\Pi$ denotes the distributional Bellman operator defined in Section 2.
>
> **6.** Before Sec 3.4:
> Is Relu is a good choice? What is your thought on other functions? How to choose g in practice?
> The prediction error would be high ->The prediction error would be higher
> eq 11: This is similar to Mavrin's idea: using exploration bonus -- UCT style.
>
> **Response**: Thank you for your comment. As for the choice of function $g$, ReLU ensure the output range of $g$ to be $[0,\infty)$. Some other function is also applicable, we may design some experiments to compare the performance in future. Some other minor issues are addressed in the revised paper.

---

> ### Author Response · Authors · 2020-11-25
> **Reference**
>
> [1] Will Dabney, Mark Rowland, Marc G Bellemare, and R ́emi Munos. Distributional reinforcement learning with quantile regression. In AAAI, 2018a.
>
> [2] Will Dabney, Georg Ostrovski, David Silver, and Remi Munos. Implicit quantile networks for distributional reinforcement learning. In International Conference on Machine Learning, pp.1096–1105, 2018b.
>
> [3] Derek Yang, Li Zhao, Zichuan Lin, Tao Qin, Jiang Bian, and Tie-Yan Liu. Fully parameterized quantile function for distributional reinforcement learning.  In Advances in Neural Information Processing Systems, pp. 6193–6202, 2019.
>
> [4] Borislav Mavrin, Shangtong Zhang, Hengshuai Yao, Linglong Kong, Kaiwen Wu, and Yaoliang Yu. Distributional reinforcement learning for efficient exploration. In International Conference on Machine Learning, pp. 4424–4434, 2019.

---

### Decision · Program_Chairs · 2021-01-07
**Final Decision**

**Decision:**

Reject

**Comment:**

This work proposes a non-decreasing quantile functional form for distributional RL, and secondly propose using the distributional error as a means of exploration. The experimental results are very exciting. The paper, however, needs further work before acceptance: the reviewers raised concerns about Theorem 1: a full proof is not included (nor written convincingly during discussion), and while several encouraging experiments were added during the discussion to the paper addressing the reviewers concerns, they fell short (understandably, given the time available).

Thus on this basis, I recommend rejection at this time, but think it likely that with these adjustments the paper will be accepted in future.